# In Pursuit of a Comprehensive Understanding of Expertise Development: A Comparison between Paths to World-Class Performance in Complex Technical vs. Endurance Demanding Sports

**DOI:** 10.3390/sports10020016

**Published:** 2022-01-28

**Authors:** Martine Aalberg, Truls Valland Roaas, Morten Andreas Aune, Øyvind Bjerke, Tore Kristian Aune

**Affiliations:** 1Department of Sport Science, Sport and Human Movement Science Research Group (SaHMS), Nord University, 7600 Levanger, Norway; martine_aalberg@hotmail.com (M.A.); truls.v.roaas@nord.no (T.V.R.); morten.a.aune@nord.no (M.A.A.); 2Department of Teacher Education, Norwegian University of Science and Technology, 7491 Trondheim, Norway; oyvind.bjerke@ntnu.no

**Keywords:** expertise, specialization, freeskiing, cross-country skiing, deliberate play, deliberate practice

## Abstract

A comprehensive understanding of skill acquisition is important for different performance domains, and has practical implications for both sport sciences and public health. The study compared important constraints for expertise development in a physically demanding sport (cross-country skiing) versus a technically demanding sport (freeskiing). Eighteen world-class athletes reported the importance of different constraints for their developmental history subdivided into two age spans: (1) 7–15 years and (2) 16 years until present. The total amount of training did not differ between the groups, but from the age of 16, the cross-country skiers spend approximately 98% of their training specific to their main sport, compared to 75% for freeskiers. No differences were found between the distribution of organized versus non-organized training in main sport, but freeskiers reported a higher amount of unorganized training in other sports after the age of 16. No differences were found in perceived importance of facilities, enjoyment of performing their sport, or the need for early specialization of training. After the age of 16, the cross-country skiers reported a higher need for coach involvement compared to freeskiers. The two sports mainly share common paths to expertise but differ in the need for specific training and coach involvement.

## 1. Introduction

A comprehensive understanding of the complexity of skill acquisition is important for different performance domains of all ages and tasks and has practical implications for both sport sciences and public health (e.g., lifespan skill development, physical education, rehabilitation, elite sport). Studies of paths to high performance in different domains have the potential to elucidate and rate knowledge about significant constraints for the development of specific expertise, and for the improvement in skills in general. Over the recent decades, there has been an increase in expert performance development, in general, and, in particular, within sports performance. Expertise development in sport is a result of an arduous process that emerges as a result of interactions between multiple sport-specific constraints. There seem to be different pathways concerning how this expertise is accomplished and how different athletes achieve gold medals in Olympic sports [1]. Researchers have mainly pointed out two different pathways to achieve expert performance in sports [2]. The pathways are usually described as either a coach-led and highly sport-specific way termed “deliberate practice” or as a non-organized (peer-led) activity termed “deliberate play” [1].

In sports where peak performance is attained before puberty (e.g., women’s gymnastics, figure skating), early specialization is often required to reach elite performance; however, in sports where athletes reach peak performance in their late 20s or early 30s, early diversification is described as the best pathway to expertise [2]. While there are several definitions of expertise, sport performance expertise is defined as the ability to consistently demonstrate superior athletic performance [3].

Expertise in sports is a result of interacting constraints; these different constraints are described as the numerous variables that form the expert’s developmental trajectory [4]. The literature about expert performance development describes several constraints that are considered to be crucial to reach a level of expertise within a field. A crucial constraint that has been studied to a large extent is the amount of time spent on one’s main sport. Simon and Chase [5] estimated that expert chess players had spent approximately 10,000 to 50,000 h playing or studying chess. They also found that a minimum of 10 years of training is necessary to reach an expert level, which is also revealed in the power law of practice by Newell and Rosenbloom [6]. The “10-year rule” has been widely used in the sports domain [7,8]. The validity of the “10,000-h rule” has been investigated [9,10,11], and the main critique is the lack of measures for variation in the amount of training leading to expertise. Gobet and Campitelli [10] studied the variation in the number of training hours required to reach the level of expertise and found that chess players spent between 3000 and 23,000 h of training to reach expertise, with the average number of hours being 11,000. Gobet and Campitelli [10] concluded that the number of hours with domain-specific training could not be the only requirement for achieving expertise. In the context of Olympic gymnastics, research found that the athletes had accumulated 18,835 training hours by the age of 16 [12]. The key point is diversity in accumulated training hours to reach expertise in different sports and for individual athletes, as shown by Güllich [13].

Another interesting constraint for expertise development is whether the athletes are a result of early specialization or early diversification. The ongoing debate is whether athletes become experts through early specialization in their main sport or through a varied and diverse road of experiences before specialization at a later stage [13]. There is no consensus in the definition of specialization in sports [14]; but it refers to athletes who focus on one sport and on specific training, often defined as intensive year-round training in one sport to the exclusion of other sports, participation in the main sport at least eight months of a year, or the termination of practice of all other sports to focus on the main sport [15,16]. This corresponds to the widely used conceptual term *deliberate practice* [7], where frequent repetitions and coach corrections in training are made, often through a monotone repetition in a specific context. Ericsson et al. [7] claimed that practice should be domain specific, and that individuals who started specializing early would always perform better than those who started late. This is exemplified with expert musicians, figure skaters, and soccer players who began practice at five years of age [8,9,17]. Deliberate practice is described as structured and organized activity that requires cognitive or physical effort with the primary goal of improving an important aspect of current performance [8]. Through coach-led organized training, the intention is to improve previous performance and deliberate efforts to change particular aspects of performance without immediate reward. In sports where peak performance is attained before puberty (e.g., women´s gymnastics, figure skating), early specialization is often required to reach elite performance [12]. The study by Law et al. [12] on rhythmic gymnastics demonstrated that Olympic gymnasts participated in fewer than two additional activities from age 4 to 16, stating that early specialization in their main sport is necessary to reach a world-class level.

A contrasting pathway to expertise is early diversification. Early diversification refers to reduced early sport-specific training and, instead, experience throughout various sports experiences that is hypothesized to facilitate later performance (see Güllich, [1]) for a more detailed description). Early diversification is related to the conceptual framework of deliberate play, which builds on the idea that sports activities are non-organized and self-regulated and that the athletes themselves are interested in discovering and exploring activities that are peer-led and without coach involvement [18]. Several studies have argued that deliberate play is important to improving motor skills, such as skill acquisition in Brazilian soccer players [19], or tactical creativity and tactical intelligence for basketball players [20].

Some evidence from team sports suggests that a considerable number of training hours in both deliberate practice and deliberate play can contribute significantly to the development of expertise in sports [21,22]. Furthermore, Soberlak and Côté [22] found that elite ice hockey players spent the same amount of time in deliberate play activities as deliberate practice activities before the age of 20. Using an ecological approach [23,24], sports practitioners can be seen as landscape designers for learning, shaping an environment, and adopting tasks to explore learning. With an ecological approach, the athletes are wayfinders that individually learn to self-regulate through unfamiliar landscapes in a skillful way. However, independent of categorization, training specificity is crucial for performance enhancement [25]. Presumably, there are several contents that are valuable for athletes, as Güllich [13] proclaimed, there are many roads that lead to Rome.

Research has shown that expertise in sports develops through interactions between an individual and their performance environment [26,27]. To understand pathways towards expertise, it is interesting to evaluate how the athletes themselves perceive the importance of different constraints on their expertise development. Constraints such as (1) physical facilities (access to training facilities and equipment) [28], (2) coach involvement [29,30], (3) athletes needing to be part of an organized group [31,32], (4) athletes feeling the enjoyment of training and competition [28,33], and (5) the need for specialization in their main sport [12,34].

The purpose of the present study is to examine potential similarities and differences in pathways to skill development and world-class expertise in complex technically versus physically demanding sports: (1) Freeski, which is regarded as a high demand of coordinative and technical skills, accompanied by courage [35], compared to (2) Cross-country skiing which is a very physically demanding sport that mainly requires high endurance capacity, high peak oxygen uptake (VO_2PEAK_), movement-specific strength, and only a few repetitive skiing techniques [36,37,38].

As freeski is a relatively new discipline in alpine sports; there is limited research on how expert performance is acquired. Freeskiing has two subgenres: freeriding, otherwise known as big mountain skiing, which refers to skiing extremely steep runs off-piste and on exposed terrain; and freestyling, which consists of performing aerial tricks using kickers, half-pipes, or other obstacles. Performance in competition is scored on dynamical criteria such as progression, amplitude, variety, execution, and difficulty. Sports that require complex coordinative skills and techniques (e.g., gymnastics) have firm traditions of early specialization and coach-led practice. As peak performance is reached at an early age, there might be reasons to consider that this is the case for freeskiers, as well. There are similarities in the requirements for complex coordinative skills and techniques between freeskiers and gymnasts. However, a common perception is that freeskiers, as the term implies, are less or not dependent on coach-led activity.

In comparison, cross-country skiing is regarded as a very physically demanding sport where high-performance athletes are associated with high VO_2max_ [36,39,40,41]. Cross-country skiing has normally focused on the periodization of different training methods [42], different physiological [39,43], and biomechanical constraints [44], or a combination of the above [45]. Due to the high physical demands of cross-country skiing, young cross-country skiers normally follow a traditional coach-led and specified training regime influenced by prescriptive “best practices” [40,46].

### Study Purpose

Based on the presented considerations, the aim of the current study was to investigate the impact of significant constraints on expert developmental history for world-class freeride skiers compared to cross-country skiers. Potential similarities and differences in the two groups on the road to expertise are described throughout the following subcategories of constraints:Comparison of the total number of hours of training.Early specialization versus early diversification: Comparison of time spent on main sport versus other sports.Comparison of the distribution of organized versus non-organized training.Comparison of the athletes’ perceived importance of different constraints on expertise development.

## 2. Materials and Methods

Two groups of world-class winter athletes participated in this study: (1) freeskiers and (2) cross-country skiers. The selection criteria were that athletes had to be competing regularly at the highest level in their sport (e.g., World Cup, World Championship, Olympic Games, Freeride World Tour). The sample included 8 freeskiers (three females and five males) and 10 cross-country skiers (eight females and two males), 18 participants in total. See Table 1.

The participants were recruited by personal communication, and the questionnaire was sent to the participants by e-mail or social media. All participants received written consent to sign if they wanted to participate in the study. The participants were informed of the procedures, including information about the voluntary nature of participation and the possibility to withdraw from the study at any time without giving any reasons or facing any consequences. All participants received a survey link to a digital questionnaire. All participants provided written consent to participate in the study. The study was conducted according to the guidelines of the Declaration of Helsinki, and approved by the Norwegian Social Science Data Services (NSD) with reference number 502539.

The questionnaire was adapted from previous research [9,47,48] with some additional questions and customized adjustments. The study by Hodges and Starkes [47] examined athletes’ training history by asking the athletes to recall their practice and play throughout their career and, likewise, examined the relevance of different activities. In addition, some customized adjustments were made in the questionnaire by adding questions about the impact of other significant constraints. These new constraints were the athletes’ perceived importance of facilities, coach involvement, being part of an organized group, enjoyment, and specialization, which were not investigated in Helsen et al. [9], Hodges and Starkes [47], or Hodges et al. [48]. Due to the lack of research on athletes’ perceived importance, new information and knowledge will be vital for a better understanding of athletes’ expert development. A 5-point Likert-type scale was used to assess the athletes’ ratings of the importance of the respective constraints.

### 2.1. Questionnaire

The first section of the questionnaire asked for biographical information about the athlete: (a) level of competition, (b) the age they initiated both organized and unorganized training, and (c) the age they began systematic training in their main sport. The second section asked for information about the athletes’ training history. The training history was subdivided into two age spans: (1) 7–15 years and (2) 16 until the present. In both age spans, the subjects were required to estimate the number of hours of training in: (a) organized and unorganized training in their main sport, (b) organized and unorganized training in other sports, (c) training alone in their main sport, and (d) time spent in other activities in a typical week at this age. In the third and last part of the questionnaire, the athletes had to rate their perceived importance of different constraints. The questionnaire took approximately 15 min to complete.

### 2.2. Data Analysis

Athletes’ recall of training volume was reported as hours of training during a year from the age of seven until the date of the survey. Mean training time in organized and unorganized training, as well as hours of training alone in main sport and other sports in both age groups, was found by multiplying the mean weekly time in practice by weeks within a year. This was later converted to percentages and used to analyze differences between the two groups.

Welch’s unequal variances *t*-tests were used to determine differences between groups (freeskiers versus cross-country skiers), while dependent t-tests were used to determine differences between age categories in the respective sports. For independent *t*-tests, Cohen’s (d_z_) was applied as a measure of effect size [49,50,51], in which the criteria to interpret the magnitude of the ES were: 0.0–0.2 trivial, 0.2–0.6 small, 0.6–1.2 moderate, 1.2–2.0 large, and >2.0 very large [50,52]. A 95% confidence interval (CI) on the effect size between groups was used. All statistical calculations were performed with Predictive Analytics Software (PASW, IBM, US; previously SPSS) Version 26.0 with an alpha level of significance set at *p* = 0.05 as the criterion for statistical significance.

## 3. Results

Figure 1 reports mean accumulated training hours for both sports and age groups from the age of seven until the present day. The overall results showed no significant difference in mean accumulated training hours between the freeskiers and the cross-country skiers, except at the age of seven [t(16) = 2.79, *p* = 0.039, d_z_ = 1.32 (95% CI [0.72, 2.34])]. In total, the mean accumulated hours of training from seven years of age until the time of the survey is 12,162 h for the freeskiers and 10,698 h for the cross-country skiers.

### 3.1. Content and Organization of Training

Table 2 reports accumulated training for the freeskiers and the cross-country skiers for each age span. Between 7 and 15 years of age, the main sport accounted for approximately 63% of accumulated training hours for both freeskiers and cross-country skiers, while other sports accounted for the remaining 37% of the total training hours. From age 16 years until the time of the survey, there is a significant difference in the distribution total amount of training between the main sport and other sports for the freeskiers compared to the cross-country skiers. The freeskiers spend 75% of their total amount of training time in their main sport, while the cross-country skiers spend up to 98% of their total accumulated training time [t(16) = 3.94, *p* = 0.009, d_z_ = 01.87, (95% CI [0.72, 2.98])] in their main sport.

In Table 3, accumulated hours of training are divided between organized versus non-organized training. No significant differences were found between the freeskiers and the cross-country skiers in the distribution of organized versus non-organized training in the span of 7–15 years of age. In the age span from 16 years until the time of the survey, there is a significant difference in the distribution of organized versus non-organized training between the freeskiers and the cross-country skiers. The freeskiers spend 24% of their training in other non-organized sports, compared to only 1.8% for the cross-country skiers [t(16) = 4.05, *p* = 0.008, d_z_ = 1.92, (95% CI [0.76, 3.04])].

### 3.2. The Impact of Athletes’ Perceived Importance of Different Constraints on Expertise Development

Evaluation of the athletes’ perceived importance of various constraints is presented in Table 4. The results showed no significant difference in how the freeskiers and cross-country skiers rate the importance of the respective constraints in the age span of 7–15 years. Nevertheless, the results show that both coach involvement and being part of an organized group become more important for the cross-country skiers throughout their careers (from the age of 16 years until the present) compared to the freeskiers, [t(16) = 3.74, *p* = 0.006, d_z_ = 1.79, (95% CI [0.65, 2.87])] and [t(16) = 4.48, *p* = 0.001, d_z_ = 2.13, (95% CI [0.95, 3.29])], respectively.

## 4. Discussion

The purpose of the present study was to examine potential similarities and differences in pathways to skill development of complex technically versus physically demanding sports in general, and the impact of significant constraints on expert development for world-class freeskiers compared to cross-country skiers in particular.

Essential constraints for expert development are the amount of training and the distribution of the type of training. While the results showed no difference in the total amount of training between the cross-country skiers and freeskiers, the distribution of the amount of training between the main sport and other sports differentiated. From the age of 16 onward, the cross-country skiers define approximately 98% of their training as specific to their main sport, compared to 75% for the freeskiers. No differences were found between the distribution of organized versus non-organized training for the cross-country skiers versus freeskiers in their main sport, but the freeskiers reported that they continued after the age of 16 with a significantly higher amount of unorganized training in other sports. In addition, the present study also evaluated how the athletes perceive the importance of significant environmental, organismic, and task constraints for their expert development. No differences were found for how the athletes perceived the importance of facilities, enjoyment of performing their sport, or the need for specialization, but interestingly, after the age of 16, the cross-country skiers reported a significantly higher need for coach involvement for their expertise development compared to the freeskiers.

The world-class cross-country skiers and freeskiers have engaged on average between 10,000 and 12,000 h of training, respectively. The present results are consistent with former research on the number of hours of training expert athletes invest towards their status as experts, suggesting a relationship between investment in training hours and expert performance [2,22,52]. Previous studies in cross-country skiing have shown an accumulation of 750–900 h of training per year [39,53], and, in total, an accumulation of 10,709 h from the age of 13 to 30. The most successful Olympic cross-country skier accumulated 14,300 h of training throughout her career [42]. While there are no earlier studies of freeskiers regarding the amount of training, the present results are in line with what is observed in adolescent alpine skiing and Olympic gymnasts [12,54].

While hours of training is a constraint for performance development towards expertise, the discussions about the variation in the amount of training leading to expertise could be more nuanced [13]. It is necessary to underline that several studies have shown a variation in accumulated training hours in the development of expert performance [10,11,12]. For example, Gobet and Campitelli [10] found large differences in the number of training hours required to reach the level of expertise, indicating that chess players spent between 3000 and 23,000 h of training to reach expertise. It is also necessary to emphasize that even relatively equal numbers of training hours do not ensure equal performance development, highlighting that the athletes’ performance might unfold at individual and uneven speeds. There has been a general myth about differences between sport disciplines that are seemingly more-or-less organized and the amount of training necessary to reach expertise (i.e., that traditional organized endurance sports, such as cross-country skiing, are more demanding regarding training hours than unorganized sports, such as freeskiing). The present study challenges this point of view, and the results busted this myth. The key point must be individualized diversity in accumulated training hours to reach expertise, as shown by Güllich [13].

The present results show no difference in mean age between freeskiers and cross-country skiers regarding the age they started organized training in their main sport, respectively, 11.3 and 9.1 years of age. Both groups started remarkably later compared to what previous expertise research has reported, i.e., expert musicians, figure skaters and soccer players tend to start organized training at five years of age [7,9,17]. In addition, there is no difference between freeskiers and cross-country skiers in the distribution of training between their main sport and other sports in the age span of 7 and 15 years. Between 7 and 15 years of age, the main sport accounted for approximately 63% of the accumulated training hours for both freeskiers and cross-country skiers, while other sports accounted for the remaining 37% of total training hours. From the age of 16 onward, there is a significant difference in the distribution of the total amount of training between the main sport and other sports for freeskiers compared to cross-country skiers. The freeskiers spend 75% of their total amount of training in their main sport, while the cross-country skiers define as much as 98% of the total accumulated training in their main sport. Based on these findings, both sports consider specialization as a necessity for expertise development after the age of 16. However, Gobet and Campitelli [10] concluded that the number of hours with domain-specific training alone cannot be the only requirement for achieving expertise. Based on the current results, or former research as such, it is by no means possible to conclude whether expertise development in either freeskiing or cross-country skiing is dependent on early diversification or early specialization [14]. However, the results are supported by previous research, suggesting that training hours related to the conceptual terms deliberate practice and deliberate play contribute positively to expertise development [21,22].

The third aim of the present study was to compare differences in the organization of training as a constraint for expert development. The distribution of organized versus non-organized training for cross-country skiers versus freeskiers in the main sport was not significant, but the freeskiers reported that they continued after the age of 16 with a significantly higher amount of unorganized training in other sports. The results showed that freeskiers and cross-country skiers started participating in unorganized training in their main sport earlier than organized training. Freeskiers started participating in unorganized training at 10 years of age, while cross-country skiers started at 7.3 years of age. A plausible explanation for this is the sporting culture related to winter sports, with an unorganized entrance to training reinforced by family influence [31,55,56]. No significant differences were found between groups in their time spent in organized or unorganized training between 7 to 15 years of age, and both groups reported no significant difference in the number of hours in organized versus non-organized training during this age span. This is surprising, considering the nature of the two sports, i.e., cross-country skiing is seen as a highly organized sport and freeskiing is seen as an unorganized sport. However, a significant difference between freeskiers and cross-country skiers was found in the number of hours of involvement in non-organized sports for the age span of 16 and older. A plausible explanation for this may be found in the cultures and traditions of the two sports. Freeskiing is a relatively new discipline and is apparently open and less prescriptive regarding development. However, cross country skiing has strong traditions regarding training and best practices in the sport, as seen through the prescriptive developmental stages proposed by the Norwegian Ski Federation [31,53,57].

The fourth aim of the present study was to compare the athletes’ perceived importance of the different constraints on expertise development. No significant differences were found between freeskiers and cross-country skiers in their perceived importance of facilities. Although the freeskiers rated the importance of facilities considerably higher than cross-country skiers between 7–15 years of age. It is interesting to note that freeskiers perceive the importance of facilities similarly in both age spans, while cross-country skiers viewed facilities to be of increasingly higher importance with age. The significant increase in training hours in the main sport for cross-country skiers might be an explanation of why facilities are perceived to be of greater importance above 16 years of age. The present results are in accordance with previous research and highlight the importance of natural accessibility to facilities for expert development [58,59].

There is no significant difference in rating between the perceived importance of coach involvement for freeskiers and cross-country skiers at the age of 7–15 years, but a significant difference was found between the two sports in the age span of 16 years and older. Freeskiers rate the importance of coach involvement below average, while cross-country skiers rate coach involvement as very important. These findings support the notion about freeskiers with a parallel to their involvement in non-organized activities (discovering their own pathway to expertise) and in contrast with the cross-country skiers’ organized and prescriptive approach, making the need for coach involvement very different. The present results for cross-country skiers are in line with former research, showing that coach involvement is perceived to be necessary for expertise development as athletes grow older [60]. This could also be related to findings from Ericsson et al. [61] about deliberate practice, where coaches seemingly have the ability to maximize training time, make the “right” choices, prioritize what has to be improved, and create an optimal training environment for the athlete. The contextual differences can pinpoint an essential nuance to former research about coach involvement for expertise development.

It could be unnecessary to compare how freeskiers and cross-country skiers perceive the importance of being part of an organized group to develop expertise, since they are individual athletes. However, former research has shown that being part of a performance group with teammates is important for the development of individual athletes [31]. The present results show that cross-country skiers above 16 years of age perceived being part of an organized group as significantly more important compared to freeskiers. The cross-country skiers’ ratings of being part of an organized group are supported by other studies [31,32], which argue that teammates are important to ensure quality in training, increase the effect of matching in training, and exchange experiences and knowledge. The findings are in line with previous studies in cross-country skiing, showing that former Olympic champions Thomas Alsgaard and Marit Bjørgen have emphasized the importance of discussions with teammates for their expertise development as cross-country skiers [42,62]. In contrast, freeskiers deviate from traditional rule-bound sport cultures because skill development is often conducted by individual discovery learning with subcultural similarities found in other action sports [63]. Due to these factors, these athletes place less importance on being part of an organized group for their expertise development. Freeskiing athletes have, from the historical beginning of the sport, manipulated task constraints (i.e., tricks, jumps, and other features in the environment) to create movement challenges and overload, resulting in the need for teammates, strict organization, coach instructions, and assessment to be unessential by design.

Both freeskiers and cross-country skiers have spent a substantial number of hours in organized training throughout their careers, and both groups rate the importance of organized training as average for their expertise development. Both groups consider organized training to be more important in early age (between 7–15 years of age) compared to older age (16 years and older), but no significant differences between groups were found. In addition, freeskiers also reported a high importance of extensive unorganized training, and confirmed their preference of independence, use of discovery learning, and deliberate play. This is in accordance with Ryan and Deci [64], who emphasized the athletes’ desire to initiate their own actions, e.g., skiing individual lines off-piste or jumping over obstacles that are mostly prepared by themselves [65]. In addition, cross-country skiers perceive unorganized training as important, which might be interpreted as conflicting since they perceive coach involvement and being part of an organized group as important, as well. This might be explained by the governing policies in cross-country skiing that, on the one hand, recommend and encourage the development of independent athletes [57] but, on the other hand, have an indoctrinated system to emphasize the importance of best-organized practices and the significance of coaches. This will possibly lead athletes to find support and knowledge from coaches if they are seen to have a vital role in the planning and evaluation of training [42], but then execute the training without the coach present, making the appearance of training to be unorganized when training alone.

Acquisition of expertise requires athletes to invest time and effort, as well as have the passion and desire to improve performance. A collective term in this matter is enjoyment, which is seen as fundamental for prolonged activity, no matter the context. Both freeskiers and cross-country skiers rate enjoyment as highly important for their expertise development, and the results showed no significant differences in the rating of enjoyment between the two groups at either age span. High ratings for enjoyment at an early age can be an important factor for sport participation later in their career [56]. Nevertheless, nuances in the factors behind individual enjoyment must not be underestimated. In the original work of Ericsson et al. [7], deliberate practice towards expertise was not inherently enjoyable. In light of this alleged notion, a separation between practice that was high on relevance but low on enjoyment was put forward. A priori, this separation is speculative. Research has shown that when consequences of an activity are differentiated from its inherent enjoyment, enjoyment ratings are generally lower, yet still relatively high [48,66]. Studies have shown that demanding practice activities with high relevance can be enjoyable [9,47,66,67]. The collective term enjoyment could be viewed as polysemic because there is a range of causes behind athletes’ personal preferences regarding enjoyment, and future research should nuance the multifaced nature of individual enjoyment.

Whether athletes become experts through early specialization in their main sport or through early diversification with various experiences before specialization at a later stage remains ambiguous. Neither freeskiers nor cross-country skiers reported that they perceive early specialization as particularly important between 7–15 years of age, but the results show an increase in the perceived importance of specialization in their main sport above the age of 16. No significant differences concerning the perceived importance of early specialization were found between freeskiers and cross-country skiers at either age span. The increase in the perceived importance of specialization combined with an increase in the number of training hours in their main sport is supported by previous studies [28,68], which found that athletes gradually specialized throughout their career. The ongoing debate of whether early specialization or early diversification is best for athletes is futile. Arguments are pervaded with agents’ perspectives and confirmation biases concerning advantages. With certainty and through practical experience, it can be proclaimed that it is possible for athletes to reach an expert level in sports with either early specialization or diversification in the early stages of their sporting careers [69,70].

### Practical Implications for Researchers, Coaches and Athletes

The results from the present study have to be interpreted according to the modest sample size. The total *n* is a consequence of examining the importance of different constraints in world-class freeski and cross-country skiing athletes, and when the inclusion criteria are restricted to top-ranked elite-level performers the total sample size has to be relatively low. Based upon the nature of expertise, the key practical implication emphasized to develop expertise is individualization, not generalization. Hence, there are some logical practical implications related to the present findings. Firstly, the problem when describing and comparing differences in the organization of training is the unnatural polarization of categorization. Researchers often attempt to place descriptive data in a distinct dichotomy manner, which leads to conclusive speculative “evidence” for athletes’ pathways to expertise. Based upon the present results, the invented borders between play and practice are merely fulfilling a need to categorize training when it comes to expertise development, instead of providing nuanced directives based upon specific principles. Both coach-led and unorganized regimes could be helpful for expertise development, instead of biasing one side, researchers should recognize the individuality of an athlete’s path to expertise. Observational, empirical, and longitudinal data can contribute to supporting the principle of individuality. It is perhaps time for researchers to change some of the terms used to describe experts’ developmental pathways since the unsubtle conceptual frameworks of deliberate practice and play often act as an argument for “what to do”. In order to reach world-class performance, there are numerous factors involved, but in general, everyone performing at the highest level has spent a tremendous number of hours training the sport they are experts in (specificity). Thus, for coaches, the principle of specificity in combination with a continuous increase in task and environmental complexity in an individualized setting is crucial in constructing representative learning conditions. The awareness for coaches when athletes make the transition into specializing and extensive training is also to “take the hand of” when the time is right in order to give the athlete self-awareness and allow them to be more self-regulated when exploring and discovering movement patterns and possibilities (individualization). Perhaps athletes should participate in more unfolding self-initiated training and positive competition with others—facilitating subjective enjoyment components. We proclaim that the multitude of different ways to organize training may not be captured in single terms. However, as world-class freeskiers reported, specificity is a crucial constraint to achieve expert performance, although organized training and coach involvement were less decisive constraints to attain expert performance from the age of 16. As a consequence, it is interesting to discuss whether the term unorganized training should be shelved or renamed with the more nuanced term self-organized training, and as a consequence, researchers, coaches, and athletes have to define what self-organize is in their particular context.

## 5. Conclusions

The purpose of the present study was to compare the similarities and differences of the important constraints on the development of expertise for world-class freeskiers versus cross-country skiers. In conclusion, freeskiers (complex technically demanding sport) and cross-country skiers (very physically demanding sport) in general share common elements and roads to expertise. However, interestingly, the observed higher need for specific training and coach involvement for cross-country skiers compared to freeskiers creates new questions for future research in the search for arbitrative constraints for specific expertise, and for skill development in general.

## Figures and Tables

**Figure 1 sports-10-00016-f001:**
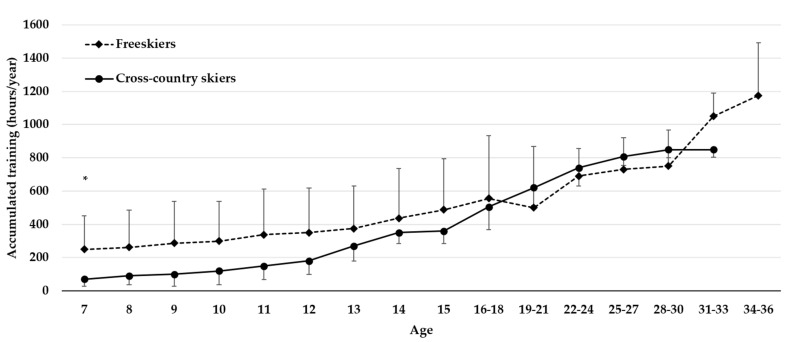
Total number (mean and standard deviation) of accumulated training hours from the age of seven until the time of the survey. * Indicates a significant difference between freeskiers and cross-country skiers.

**Table 1 sports-10-00016-t001:** Descriptive statistic for biographic information of the athletes in each sport.

Group	Freeriders (N = 8)			Cross-Country Skiers (N = 10)		
	Minimun	Maximum	Mean (SD)	Minimun	Maximum	Mean (SD)
Age range of the participants	19	37	27.5 (7.1)	22	32	27.5 (3.4)
Age for first organized training in main sport	6	16	11.3 (4.2)	6	16	9.1 (3.1)
Age for first unorganized training in main sport	4	16	10 (5.1)	4	15	7.3 (4.2)
Start age for systematic training in main sport	13	16	15 (1.2)	12	16	14.6 (1.3)

**Table 2 sports-10-00016-t002:** Comparison of the distribution of total amount of training in main sport (specific training in main sport) vs. other sports for cross-country skiers and freeskiers. * Indicates a significant difference between freeskiers and cross-country skiers.

		Freeskiers 7–15 Years	Cross-Country Skiers 7–15 Years	Freeskiers 16+ Years	Cross-Country Skiers 16+ Years
		% (SD)	% (SD)	% (SD)	% (SD)
Main sport	% distribution of hours of training in main sport	63% (17.88)	63% (12.40)	75% (18.77)	98% * (2.85)
Other sports	% distribution of hours of training in other sport	37% (17.87)	37% (12.27)	25% (18.77)	2% * (2.85)

**Table 3 sports-10-00016-t003:** Comparison of the distribution of total number of hours of organized vs. unorganized training between main sport and other sports for cross-country skiers and freeskiers. * Indicates a significant difference between freeskiers and cross-country skiers.

		Freeskiers 7–15 Years	Cross-Country Skiers 7–15 Years	Freeskiers 16+ Years	Cross-Country Skiers 16+ Years
		% (SD)	% (SD)	% (SD)	% (SD)
Organized training	% distribution of hours of organized training in main sport	21.9% (17.1)	29.1% (22.8)	9.8% (0) ^†^	17.5% (49.0)
	% distribution of hours of organized training alone in main sport	24.4% (29.4)	16.3% (14.9)	23.4% (36.4)	30.1% (27.3)
Unorganized training	% distribution of hours of unorganized training in main sport	28.2% (31)	31.2% (37.2)	42.8% (15.2)	50.7% (14.4)
	% distribution of hours of involvement in other unorganized sports	25.5% (22.5)	23.3% (25.2)	24.0% * (48.4)	1.8% (9.3)

Note: ^†^ explaination of SD = 0; because only one athlete reported practicing the present type of training.

**Table 4 sports-10-00016-t004:** Freeskiers vs. cross-country skiers perceived importance of different constraints upon expertise development. * Indicates a significant difference between freeskiers compared to cross-country skiers.

Constraint	Freeskiers 7–15 Years	Cross-Country Skiers 7–15 Years	Freeskiers 16+ Years	Cross-Country Skiers 16+ Years
	Mean (SD)	Mean (SD)	Mean (SD)	Mean (SD)
Facilities	4.0 (1.31)	2.9 (1.10)	4 (1.31)	4.1 (0.57)
Coach involvement	3.25 (1.04)	3.9 (0.74)	2.25 (1.28)	4.0 * (0.67)
Being part of organized group	ND	ND	2.13 (1.13)	4.1 * (0.74)
Enjoyment	4.13 (0.64)	4 (1.05)	4.25 (0.46)	3.9 (0.88)
Specialization	2.25 (1.58)	2.2 (0.79)	3.75 (0.71)	4.1 (0.57)

Notes: 0 = low; 5 = high. No data (ND).

## Data Availability

The data presented are available on request from the corresponding author. The data are not publicly available due to rules of Norwegian Center for Research Data.

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
