# Peer review of "In Pursuit of a Comprehensive Understanding of Expertise Development: A Comparison between Paths to World-Class Performance in Complex Technical vs. Endurance Demanding Sports"

_sports, 2022, doi:10.3390/sports10020016_

Round 1
Reviewer 1 Report
I enjoyed reading your manuscript. My comments are more technical in nature.
Line 43, it seems a new paragraph here would help.
Line 121, I sub heading of Study Purpose would help the reader keep with your manuscript.
Line 128, there seems to be an extra space between techniques and the [
Table 1, age range, seems more informative than SD. Your note is not necessary, and you did not present the data as plus and minus.
Figure 1, this is my main issue. It seems a figure of effect size differences would be lots more informative than as presented. Or you present a Figure 2 with those data. You have lots of information and have spent lots of time on your research. I think asking for effect size values and interpretation is reasonable to help just a bit more in understanding and presenting your research.
Table 2, the note, does not make sense.
Table 3, I do not see ND in the table.
Line 281, it seems a reference is needed for the sentence concerning hours, respectively.
The discussion is well-written and informative.
Reviewer 2 Report
The manuscript adds to a topic that has seen a plethora of studies over the past decade, that describe and compare what the researchers see as the best path to expertise in diverse sports. And, as with other studies in the field, persistent conceptual and analytical issues are observed, as well the absence of practical implications.
The Deliberate Practice Theory can help to explain and sustain the road to excellence of musicians and chess players, but not of athletes. Rees and colleagues (2013) showed that a specialization pattern of super elite athletes simply does not exist.
Furthermore, early specialization is never clearly defined as the onset of organized training is always set somewhere between 6-12 years of age. What we know is that, at least in Europe, youth engage and specialize in sport alongside their attendance of compulsory school. The differences in hours of training and diversification of sports depend of personal choices and contextual factors.
The categories of deliberate practice and deliberate play are also a classificatory artifact, with little utility to a deep understanding of the athletes’ commitment to a specific sport.
The significant enjoyment is polysemic, meaning different things to different people. Without a clear definition, the researcher ignores the sense of the word for the athlete.
From an analytical point of view, with such a small sample and an expectable non normal distribution, t-rests and p-values have no explanatory power.
In the discussion, the differences observed between free-skiers and cross-country skiers are explained because sports are different, which is somehow tautologic.
Th sample is composed of super elite athletes, which is rare to achieve. But a retrospective comparison between them makes little sense.
Author Response
Please see the attachment.

This manuscript is a resubmission of an earlier submission. The following is a list of the peer review reports and author responses from that submission.